# Immunopathology of Pulmonary *Mycobacterium tuberculosis* Infection in a Humanized Mouse Model

**DOI:** 10.3390/ijms25031656

**Published:** 2024-01-29

**Authors:** Afsal Kolloli, Ranjeet Kumar, Vishwanath Venketaraman, Selvakumar Subbian

**Affiliations:** 1Public Health Research Institute, Rutgers-New Jersey Medical School, Newark, NJ 07103, USA; ak1482@njms.rutgers.edu (A.K.); rk879@njms.rutgers.edu (R.K.); 2College of Osteopathic Medicine of the Pacific, Western University of Health Sciences, Pomona, CA 91766, USA; vvenketaraman@westernu.edu

**Keywords:** tuberculosis, immune cells, inflammation, immunohistochemistry, spatial analysis, histology, animal model

## Abstract

Despite the availability of antibiotic therapy, tuberculosis (TB) is prevailing as a leading killer among human infectious diseases, which highlights the need for better intervention strategies to control TB. Several animal model systems, including mice, guinea pigs, rabbits, and non-human primates have been developed and explored to understand TB pathogenesis. Although each of these models contributes to our current understanding of host-*Mycobacterium tuberculosis* (Mtb) interactions, none of these models fully recapitulate the pathological spectrum of clinical TB seen in human patients. Recently, humanized mouse models are being developed to improvise the limitations associated with the standard mouse model of TB, including lack of necrotic caseation of granulomas, a pathological hallmark of TB in humans. However, the spatial immunopathology of pulmonary TB in humanized mice is not fully understood. In this study, using a novel humanized mouse model, we evaluated the spatial immunopathology of pulmonary Mtb infection with a low-dose inoculum. Humanized NOD/LtSscidIL2Rγ null mice containing human fetal liver, thymus, and hematopoietic CD34+ cells and treated with human cytokines were aerosol challenged to implant <50 pathogenic Mtb (low dose) in the lungs. At 2 and 4 weeks post infection, the tissue bacterial load, disease pathology, and spatial immunohistology were determined in the lungs, liver, spleen, and adipose tissue using bacteriological, histopathological, and immunohistochemical techniques. The results indicate that implantation of <50 bacteria can establish a progressive disease in the lungs that transmits to other tissues over time. The disease pathology in organs correspondingly increased with the bacterial load. A distinct spatial distribution of T cells, macrophages, and natural killer cells were noted in the lung granulomas. The kinetics of spatial immune cell distribution were consistent with the disease pathology in the lungs. Thus, the novel humanized model recapitulates several key features of human pulmonary TB granulomatous response and can be a useful preclinical tool to evaluate potential anti-TB drugs and vaccines.

## 1. Introduction

Tuberculosis (TB), a bacterial disease resulting from *Mycobacterium tuberculosis* (Mtb) infection, caused 1.3 million deaths and 10.6 million new cases among humans worldwide in 2022 [1]. In addition, one quarter of the global population have been estimated to harbor asymptomatic latent TB infection (LTBI) [2]. Although LTBI cases are asymptomatic, immune-suppressing host conditions can reactivate LTBI into symptomatic TB [3]. Thus, individuals with LTBI are potential sources of future TB cases. In addition, the co-existence of TB with other diseases, such as HIV infection and diabetes, increases the risk of mortality significantly [4]. The current therapeutic regimen for TB involves the administration of four antibiotics (isoniazid, rifampin, pyrazinamide, and ethambutol) for at least six months [1]. This multidrug treatment for prolonged periods of time has the risk of poor patient compliance, leading to sub-optimal cure and development of drug resistant TB (DR-TB) cases that pose additional constraints to efficient TB management [5]. Taken together, these reports highlight the urgent need to develop improved and effective therapeutic interventions for global TB control.

A key pathological hallmark of TB in humans is the formation of granuloma, an organized cellular structure occurring at the site of Mtb infection [6]. The dynamics of immune cell recruitment, activation, and function in the granulomas are mainly shaped by the balance in the pro- and anti-inflammatory cytokines and chemokines prevailing in each of the granulomas [6]. In general, granulomas are believed to contain Mtb within the site of infection; however, the bacteria can survive and replicate intracellularly within the infected immune cells of the granuloma [7]. Thus, the granulomas can be both beneficial as well as detrimental to the host’s protection against Mtb infection. Importantly, the intricate cellular and molecular events underlying the protective versus permissive granulomatous response during Mtb infection are not fully understood. 

Several animal models of Mtb infection have been utilized to explore the host–pathogen interactions underlying active TB and LTBI with variable levels of success in reproducing the pathophysiology of respective conditions in humans [8,9]. For example, pathogenic Mtb infection can produce caseous necrotic granulomas that undergo cavitation in non-human primate (NHP) and rabbit models [10,11]. These models can also establish spontaneous LTBI and reactivation of LTBI upon immunosuppression treatment, similar to human conditions [12,13,14]. These peculiar pathological manifestations of Mtb infection were poorly reproduced in most of the murine models [11]. However, due to the advantages of murine models, such as cost, ease of handling, etc., this model has been extensively used for studies of anti-TB drugs and vaccine discovery as well as to understand host response during Mtb infection [10,11]. Recently, to improve the pathophysiology of Mtb infection in the mouse model in the context of various clinical backgrounds, including co-infection with HIV, several humanized mouse models were created by transplanting human tissues, such as thymus, liver, and bone marrow, and/or immune cells [15,16,17,18,19,20]. 

Mtb infection in humanized mice derived from different strains of mouse, including C57BL/6, NOD, NOD-SCID/γ, and NSG (NOD/LtSscidIL2Rγnull), have been shown to possess functionally active human T and B cells, neutrophils, and macrophages, and produced several proinflammatory molecules in the lungs and other organs [16,21,22,23]. The human immune system NSG mice (HIS-NSG) displayed heterogeneity in granulomatous lesions, similar to TB patients, upon Mtb infection, and this model was evaluated as a preclinical tool for anti-TB drug discovery [16]. Similarly, a humanized mouse model developed to express the HLA-A11^+/+^ DRB1*01:01^+/+^H-2-β2m^−/−^/IAβ^−/−^ allele corresponding to the Chinese population was used to develop a novel polypeptide-based vaccine to protect against TB [22]. Importantly, the immune response developed in humanized mice upon Mtb infection and/or vaccination with BCG or CpG-C was complementing the corresponding response from the guinea pig model of Mtb infection [23]. These studies have contributed to the foundational characterization of the human immune system constituted in the humanized mouse model and highlighted the superiority of this model in understanding host–Mtb interactions. However, the kinetics of bacterial growth and disease pathology in tissues during a low-dose aerosol (<50 bacteria) Mtb infection in humanized NSG mice treated with human cytokines were not reported previously. Furthermore, the spatial distribution of immune cells and immune marker expression in the lungs of humanized mice infected with low-dose Mtb inoculum through aerosol exposure is not fully understood. 

Our primary objective in this pilot study was to characterize the spatial immunopathology of low-dose (<50 bacteria) Mtb infection in a humanized mouse model. We used a novel NOD/LtSscidIL2Rγ-null humanized mouse model to evaluate the spatial immunopathology of pulmonary Mtb infection with a low-dose infectious inoculum of bacteria. We observed distinct kinetics in the spatial distribution of T cells, macrophages, and natural killer cells in the granulomas, which is consistent with corresponding disease pathology in the lungs over time. We also determined the spatial expression of proinflammatory cytokines TNFα and IFNγ, as well as immune regulators, including inducible nitric oxide synthase-2 (NOS2), glutathione reductase (GR), and glutathione S-transferase M3 (GSTM3), all of which are involved in TB pathogenesis. Thus, our study further highlights the immune environment of a novel humanized mouse model during Mtb infection, which can be further utilized as a preclinical tool to test potential vaccines and therapeutic drugs for effective TB control.

## 2. Results

### 2.1. Validation of the Humanized Mice

In this study, we used NOD/LtSscidIL2Rγnull (NSG) mice for humanization by transplanting human fetal liver and thymus, followed by treatment with DNA-based transgene delivery system that produces human IL-3, IL-7 and GM-CSF. This approach of tissue engraftment followed by cytokine treatment is novel, since in the conventional humanization of mice the cell-growth promoting factors were constitutively and continuously expressed, as opposed to targeted and sequential delivery of selected factors in our model. The humanized mice generated by our method had a lifespan of about 30 weeks and had various subtypes of functional human CD45+ cells, which includes all the leucocytes such as T and B cells as reported previously [24,25]. To further validate this humanized model in the context of Mtb infection, which primarily targets macrophages in the lungs, we determined the expression of human and mouse macrophage cell surface markers (Appendix A). The expression of human macrophage-specific IBA-1 was profoundly expressed in the lungs of humanized mice, as opposed to feeble, background expression of the mouse macrophage-specific marker, F4/80. This observation indicates a predominant human macrophage reconstitution in the humanized mice used in this study.

### 2.2. Body Weight Changes in Mtb-Infected Humanized Mice 

In humans, active TB is associated with loss of body weight [26]. To determine whether Mtb infection affects body weight in humanized mice, we measured the animal weight from the day of infection until the experimental end point (28 days post infection). As shown in Figure 1A, Mtb-infected humanized mice started losing about 3% of their weight as early as 7 days post infection and sustained a similar low body weight until 28 days post infection (Figure 1A). These observations are consistent with reports of human patients with severe pulmonary TB [26]. 

### 2.3. Kinetics of Bacterial Growth in the Organs of Mtb-Infected Humanized Mice 

To determine the establishment of Mtb infection in various internal organs of humanized mice, we measured the bacterial load in the lungs, liver, spleen, and visceral white adipose tissue (adipose) by the CFU assay (Figure 1B). An average of 37 (±9) Mtb CFU was implanted in the mouse lungs soon after aerosol infection (T = 0). The number of bacterial CFU in the lungs gradually increased to an average of 3 log_10_ at 2 weeks and 4.1 log_10_ at 4 weeks post infection. However, a bacterial CFU of about 3 log_10_ was noted in the liver and spleen only at 4 weeks post infection; no cultivable bacteria were seen in these organs at T = 0 or 2 weeks post infection. In contrast, low numbers of bacterial CFU were observed in the adipose tissue at 2 and 4 weeks post infection; there was no striking difference in the bacterial load between these two time points (Figure 1B). Thus, at a low-dose aerosol infection, Mtb dissemination from the lungs to other internal organs occurred as early as 2 weeks (in adipose tissue) and more markedly at 4 weeks (liver, spleen, and adipose) post infection.

### 2.4. Histopathologic Changes in the Organs of Mtb-Infected Humanized Mice 

Disease pathology in Mtb-infected humanized mouse lungs, spleen, and adipose tissue was analyzed microscopically after H&E staining. The lungs of uninfected animals showed a basal level of cellular composition without any signs of inflammatory infiltrates (Figure 2A,B). Mononuclear histiocytes resembling macrophages were clearly seen, and no sign of collagen deposition and tissue remodeling or fibrosis were noticed in these uninfected lung sections (Appendix A). In contrast, at 2 weeks post infection, a moderate level of inflammatory cellular infiltration, accumulating as foci of variable sizes, was noted in the lungs (Figure 2C,D). Inflammatory foci with a mild level of fibrosis were interspersed with the presence of clear lung parenchyma in these animals (Appendix A). At 4 weeks post infection, typical granulomas, marked with severe inflammation, accompanied by immune cell infiltration, were noted in the lungs (Figure 2E,F). These lesions that occupied the majority of the lung parenchyma contained immune and non-immune cells. Alveolar wall thickening with shrinkage of the bronchioles was also noticed at this time point. Furthermore, a moderate level of interstitial fibrosis was also noticed in the granulomatous lesions (Appendix A). Taken together, the histologic analysis of Mtb-infected humanized mouse lungs showed signs of progressive disease pathology, including gradual induction of fibrosis and tissue remodeling, starting from 2 weeks, that peaked at 4 weeks post infection (Figure 2).

Similar to the lungs, the spleen of Mtb-infected humanized mice also displayed signs of pathological disease (Figure 3). While the spleen of uninfected mice had no signs of disease pathology, Mtb-infected animals showed multiple inflammatory foci of variable sizes in the spleen as early as 2 weeks post infection (Figure 3A–D). Distinct clusters of lymphocytes surrounding inflammatory infiltrates were seen in the infected spleens at this time. The severity of inflammation and the extent of tissue involvement further elevated in the spleen at 4 weeks post infection (Figure 3E,F). Larger granulomatous lesions with inflammation surrounded by lymphocytic cuffs were noticed in the spleen of humanized mice at this time point. 

Consistent with the Mtb-infected lungs and spleen, the adipose tissue of humanized mice also showed inflammatory response at 2 and 4 weeks post infection (Figure 4). The adipose tissue of uninfected mice showed no sign of inflammation and the presence of foamy macrophages (Figure 4A,B). Accumulation of more activated macrophages were noticed in the adipose tissue after 2 weeks of Mtb infection (Figure 4C,D). Severe inflammation and increased immune cell infiltration was observed in the adipose tissue at 4 weeks post infection (Figure 4E,F).

Taken together, the kinetics of histopathological analysis of lungs, spleen and adipose tissue from 2 to 4 weeks post infection suggest dissemination of Mtb infection from the lungs to the spleen and adipose tissues and establishment of disease pathology over time in the humanized mice.

### 2.5. Spatial Distribution of Immune Cells in the Lungs of Mtb-Infected Humanized Mice 

We used IHC-based microscopic analysis and cell type-specific antibodies to map the spatial distribution of various immune cells and Mtb at the site of infection (lungs) in humanized mice. Specifically, we determined the distribution of lymphocytes (CD3), NK cells (CD56), monocytic phagocytes (CD68), and myeloid-derived suppressor cells (MDSC; CD11B and CD33) in the lungs at 2 and 4 weeks post infection. As shown in Figure 5, a significantly increased distribution of lymphocytes and NK cells was noticed in the Mtb-infected lungs at 2 and 4 weeks post infection, compared to the uninfected controls. There was no significant difference in the distribution of these two cell types between 2 and 4 weeks post infection. However, a significantly increased distribution of monocytic phagocytes was noticed in the lungs only at 4 weeks post infection, compared to 2 weeks in infected and uninfected animal lungs (Figure 5). 

In addition, similar levels of lymphocytes, NK cells, and monocytic phagocytes were noted at 4 weeks post infection. However, the abundance of the NK cell population was reduced, compared to the level of lymphocytes and monocytic phagocytes at 2 weeks post infection. The MDSCs (CD11B^+^/CD33^+^ cells) have emerged as an important immunoregulatory cell type in TB pathogenesis [27,28]. We observed significantly elevated MDSCs in the Mtb-infected lungs, compared to uninfected controls at 2 and 4 weeks post infection (Figure 6). However, there was no significant difference between MDSC distribution between 2 and 4 weeks post infection. Together, the spatial immune cell distribution analysis revealed a significantly increased numbers of lymphocytes, NK cells, monocytic phagocytes, and MDSCs as early as 2 weeks post infection and sustained at higher levels at 4 weeks post infection. Fiurthermore, abundant Mtb, mostly as clusters, was present in the infected mouse lungs at 4 weeks (Appendix A). These observations are consistent with disease progression in the lungs of Mtb-infected humanized mice, as supported by the kinetics of bacterial load and histopathology in these animals.

### 2.6. Spatial Expression of Immune Activation Markers in the Lungs of Mtb-Infected Humanized Mice 

To determine the spatial expression of TNFA, IFNG, and NOS2, which are key immune activation markers involved in TB pathogenesis, we used smFISH-based microscopic analysis on the lungs of Mtb-infected humanized mice. As shown in Figure 7, a significantly increased expression of IFNG was noticed in the Mtb-infected lungs at 2 and 4 weeks post infection, compared to the uninfected controls. There was no significant difference in the expression of IFNG between 2 and 4 weeks post infection. However, a significantly increased expression of TNFA and NOS2 was noticed only at 4 weeks post infection, compared to uninfected animal lungs (Figure 7). In addition, similar levels of significantly induced expression of TNFA, IFNG, and NOS2 were noted at 4 weeks post infection, compared to the uninfected control lungs. Together, the spatial immune activation gene expression profile suggests a significantly elevated proinflammatory response in the lungs as early as 2 weeks post infection, which is sustained at higher levels at 4 weeks post infection. These observations are consistent with and supported by the data on disease progression in the lungs of Mtb-infected humanized mice, including the kinetics of bacterial load, histopathology, and immune cell distribution in these animals.

### 2.7. Spatial Expression of Antioxidant Markers in the Lungs of Mtb-Infected Humanized Mice 

Glutathione (GSH) is a key antioxidant molecule that plays important roles in TB pathogenesis in humans and model animals [29]. We used a spatial imaging technique to determine the expression level of two enzymes in the GSH pathway, namely glutathione reductase (GR) and glutathione-S-transferase = M3 (GSTM3), in the lungs of humanized mice infected with Mtb. GR is the enzyme that reduces glutathione disulfide (GSSG) into GSH, while GSTM3 absorbs reactive toxic molecules and protects the host cells. As shown in Figure 8, compared to the uninfected control group, the number of host cells expressing GR and GSTM3 was significantly downregulated in the humanized mouse lungs as early as 2 weeks post infection, and this reduced level further persisted until 4 weeks post infection (Figure 8). No significant differences in GR or GSTM3 expression levels were observed between 2 and 4 weeks post infection. 

## 3. Discussion

In this study, we documented the immunopathology of TB in a novel humanized mouse model, which reproduced several key pathological aspects seen in human TB patients. We showed that Mtb-infected humanized mice develop progressive disease in the lungs, and dissemination of the infection occurs from the lungs to the spleen and adipose tissue. Furthermore, we demonstrated the spatial expression of immune activation markers in the lungs. Taken together, these findings suggest that our novel humanized mouse model can be a valuable tool in TB research. 

Preclinical animal models contribute significantly to our understanding of the host response to Mtb infection, to develop and evaluate the immunogenicity and protective efficacy of vaccine candidates, as well as to test potential drugs to treat TB. Several animal models of TB, ranging from non-human primates to zebra fish, have been developed and utilized in TB research over many decades. However, none of the animal models can fully recapitulate all the salient pathological features seen in the spectrum of TB in human patients. Nonetheless, each of these models contributes to some aspects of TB pathogenesis, and they are useful in a particular domain of vaccine and/or drug development for TB. To improve the value of the common mouse model of TB, researchers have developed humanized mice, in which small portions of internal organs and/or cells of human origin are incorporated into the mouse organs. Indeed, a few of these humanized mouse models have been used for TB studies, reported previously [16,19,21,22]. 

In the study by Calderon et al., the humanized mice were derived from NOD/SCID/γc^null^ or NSG mice by engrafting with human fetal liver and thymus following irradiation [21]. These mice also received human CD34+ hematopoietic stem cells (HSC) intravenously after engraftment and were administered with acidified water containing antibiotics. Upon intranasal instillation of Mtb, these humanized mice displayed disease pathology and immune cell composition in the lungs similar to patients with active TB [21]. Similarly, in a study by Arrey et.al, irradiated NSG mice of 1–3 days’ age were intra-hepatically injected with CD34+ HSCs to create humanized mice [16]. Aerosol exposure of these humanized mice with Mtb resulted in a lung granulomatous response like that seen in patients with pulmonary TB [16]. Recently, a next-generation HLA-transgenic humanized mouse model (huDRAG-A2) was created and used for intranasal infection with Mtb [19]. These mice were reported to have an abundant immune cell presence and produced pulmonary granulomas that resemble those in the lungs of humans with TB [19]. 

In our study, the humanized mice were created by a novel methodology; firstly, the NSG mice were transplanted with human fetal liver and thymus after myeloablation with Busulfan treatment. This was followed by sequential injection of mice with plasmids encoding human cytokines IL-3, IL-7, and GM-CSF for efficient immune cell reconstitution [24]. Thus, our humanized mice were created with a different approach than the previously reported models; a detailed evaluation of the immune constitution of this model has been previously published [24]. Importantly, our humanized mouse model displayed the hallmark disease parameters of active TB seen in human patients, including loss of body weight that is proportional to progressive disease in the lungs, elevated bacterial load and immunopathology, and bacterial dissemination from the lungs to the liver, spleen, and adipose tissues over time. The lung granulomas displayed human immune cell constitution, comprised T cells, NK cells, macrophages, and MDSCs. These findings are consistent with and supported by the previous reports of Mtb infection in various humanized mouse models. However, ours is the first study to report the presence of MDSCs in the lung granulomas and the dissemination of infection to the adipose tissue of Mtb-infected humanized mice. Importantly, the frequency of MDSCs in the blood and lungs correlated with the severity of pulmonary TB disease in humans and Mtb-infected mice, respectively [28,30,31]. In addition, MDSCs have been shown to harbor Mtb and induce NOS2 expression, thus contributing to disease pathology [32,33]. Consistent with these reports, we observed significantly elevated MDSCs in the lungs that was proportional to the bacillary load and disease severity in the Mtb-infected humanized mice. 

Another important aspect of our study is the application of IHC (proteins/antibody) and smFISH (transcript/mRNA)-based spatial immunopathology profiling of Mtb-infected humanized mouse lungs. We observed significantly elevated expressions of TNFA, IFNG, and NOS2 in the humanized mouse lungs infected with Mtb, compared to the controls. Although IFNγ and TNFα were reported to collectively contribute to controlling Mtb growth in macrophages, these two cytokines were not considered as essential for T cell mediated protective immunity against TB [34,35]. Furthermore, significantly elevated levels of these proinflammatory molecules have been reported in the plasma of patients with drug-sensitive and drug-resistant TB, compared to the controls [36]. Together, these observations in human clinical studies and in model systems are consistent with our current observations of immune activation marker upregulation in the Mtb-infected humanized model.

GSH is an important antioxidant that plays a host-protective role against Mtb infection in humans and animal models by its direct antimicrobial as well as host immune-boosting mechanisms [29]. Indeed, significantly reduced levels of GSH were found in the plasma and immune cells of patients with pulmonary TB [37]. Furthermore, supplementation of GSH was shown to improve Mtb killing by macrophages in vitro and in model animals [38,39]. In this study, we observed a significant reduction in two GSH pathway enzymes, GR and GSTM3, in the lungs of Mtb-infected humanized mice. These results suggest that the reduction in the frequency of GR and/or GSTM3 positive cells in the lungs might lead to increased oxidative stress, which is related to the elevated bacterial load and worsening of disease pathology during Mtb infection. Therefore, consistent with human clinical studies, our humanized mouse model also suggests that GSH metabolism is impaired during Mtb infection, which might contribute to exacerbated bacterial growth and inefficient immune cell activity in controlling the infection. This is the first report showing the spatial expression of GR and GSTM3 in Mtb-infected humanized mouse lungs.

## 4. Materials and Methods

### 4.1. Mycobacterium Tuberculosis Growth Conditions

The pathogenic *Mycobacterium tuberculosis* H37Rv (Mtb) strain was originally obtained from Dr. Thomas Shinnick (Center for Disease Control (CDC), Atlanta, GA, USA). The bacteria were grown to mid-log phase (OD600 = 0.6–0.7) in Middlebrook 7H9 medium (Difco BD, Franklin Lakes, NJ, USA) supplemented with 10% ADC (albumin dextrose catalase) and stored frozen at −80 °C until ready to use. The number of bacteria in the inoculum was evaluated by serially diluting the culture in sterile 1× PBS containing 0.05% Tween-80 and plating on 7H10 agar medium supplemented with 10% OADC. The plates were incubated at 37 °C for 4–6 weeks and Mtb colonies (CFU) were enumerated. For the mouse aerosol infection, stock vials were thawed, diluted in sterile 1× PBS, and used as we described previously [13].

### 4.2. Humanization of Mice

The NSG mice were purchased from the Jackson laboratories (Bar Harbor, ME, USA). These mice were humanized as described previously [24]. Briefly, 6–8-week-old female NSG mice were treated intraperitoneally with Busulfan (Millipore-Sigma, St. Louis, MO, USA) at 30 mg/kg dose for 24 h to deplete myeloid cells. The myeloablated NSG mice were engrafted with 1–2 mm^3^ of human fetal liver and thymus in the sub-renal capsule. This was followed by intravenous injection of autologous liver-derived CD34+ hematopoietic stem cells at 1 × 10^5^ cells/mouse. To ensure and accelerate human immune cell reconstitution in the engrafted mice, plasmids encoding human IL-3, IL-7, and GM-CSF were injected intravenously, after 6–7 days post CD34+ cell injections. These plasmids produced about 23–40 pg of each of the cytokines in the reconstituted mice [24]. The presence of human immune cells was monitored in the humanized mice after 6–8 weeks by multi-color BD LSR II Analyzer-2 flow cytometry (BD Biosciences, Franklin Lakes, NJ, USA). After 8 weeks of reconstitution, the humanized mice were used for infection studies. Detailed immunological characterization of these humanized mice was previously reported [24].

### 4.3. Humanized Mice Infection and Bacterial Load Determination

Humanized mice were infected with Mtb by aerosol exposure for 40 min in a Glass-Col chamber (Glass-Col, Terre Haute, IN, USA) as described previously [39]. The bacterial inoculum for nebulization was adjusted to deliver about 50 bacteria in the lungs after aerosol exposure as we reported previously [39]. Uninfected mice were included as controls. At T = 0 (3 h post infection), 2 and 4 weeks post infection, a group of Mtb-infected mice (n = 2–3 per time point) was euthanized by cervical dislocation and necropsy was performed. Lungs, liver, spleen, and visceral white adipose tissue were harvested. A portion of these tissues was homogenized in sterile 1× PBS containing 0.05% Tween 80 and the homogenates were serially diluted and plated on Middlebrook 7H11 agar media (Difco BD, Franklin Lakes, NJ, USA). The plates were incubated at 37 °C for 4–6 weeks and the number of Mtb CFUs was counted. A portion of the tissues was fixed in 10% neutral buffered formalin for 4 days and used for histology and immunohistochemistry (IHC) studies. All experimental procedures on mice were performed in bio-safety level 3 (BSL3) containment facilities according to the approved policies and procedures of the Rutgers University Institutional Animal Care and Use Committee (RU-IACUC#PROTO201900007). 

### 4.4. Histopathological Analysis

Formalin-fixed tissues (lung, spleen, and adipose) were thoroughly washed in sterile 1× PBS and embedded in paraffin. The formalin-fixed and paraffin-embedded (FFPE) tissue blocks were cut into 5 µm slices and stained with Hematoxylin and Eosin (H&E) to visualize cellular distribution in tissues. The lung sections were also stained with Masson’s Trichrome to visualize fibrosis and tissue remodeling. The stained sections were evaluated on a Nikon Microphot DXM 1200C microscope and the images were acquired using NIS-Elements software F3.0 (Nikon Instruments Inc., Melville, NY, USA) as we previously described [13].

### 4.5. Spatial Immunohistochemistry Analysis

Spatial immunohistological analysis of lung sections was performed as we described previously [26]. Briefly, FFPE tissue sections were deparaffinized by dipping the slides with xylene for 5 min two times, followed by rehydration of the sections by washing them in graded ethanol (5 min each wash in absolute ethanol, followed by 95% and 70% ethanol). The tissue sections were soaked in a citrate buffer at 90 °C for 40 min to retrieve antigens, followed by washing in distilled water. The sections were blocked with buffer containing 5% bovine serum albumin (BSA) in 1× PBS. Solutions of anti-human CD3, CD11B, CD56, CD68, IBA-1, GR, and GSTM3 antibodies and anti-mouse F4/80 antibody were prepared in 5% BSA at 1:100 dilution, and applied to the tissue section, followed by overnight incubation at 4 °C as recommended by the manufacturer (Novus Biologicals and/or Abcam, Fremont, CA, USA). The tissue sections were washed with 1× PBS and a fluorophore-tagged secondary antibody, prepared at 1:1000 dilution in 5% BSA, was added and the sections were incubated at room temperature for 1 h. The slides were thoroughly drained and TrueBlack Lipofuschin autofluorescence quencher (Biotium Inc., Fremont, CA, USA) was applied to the tissue sections and mounted with coverslips. 

### 4.6. Spatial Gene Expression Analysis by Single-Molecule Fluorescent In Situ Hybridization (smFISH)

The smFISH technique was used to determine the spatial expression of genes encoding TNFα, IFNγ, and NOS2 on the lung sections as we previously reported [40]. Briefly, 5 µm lung sections were equilibrated in a wash buffer, containing 10% Formamide in 2× SSC, followed by overnight incubation of the sections at 37 °C in a hybridization solution containing 10% dextran sulfate, 1 mg/mL Escherichia coli tRNA, 2 mM ribonucleoside vanadyl complex (New England Biolabs, Ipswich, MA, USA), 0.02% RNase-free BSA (ThermoFisher Scientific, Waltham, MA, USA), 10% formamide, and 500 ng/mL of fluorescently-labeled probes of the target gene. After hybridization, the slides were washed twice in hybridization wash buffer at room temperature, treated with TrueBlack Lipofuschin autofluorescence quencher, and mounted with coverslips.

### 4.7. Spatial Imaging Analysis

Spatial IHC and sm-FISH images were acquired using an Axiovert 200 M inverted fluorescence microscope (Zeiss, Oberkochen, Germany) using a 20x or 63x oil-immersion objective and a Prime sCMOS camera (Photometrics, Tucson, AZ, USA) controlled by Metamorph image acquisition software (Molecular Devices, San Jose, CA, USA). To enumerate the various cell types in the lungs that express the tested target molecules (CD3, CD11B, CD56, CD68, IBA-1, GR, GSTM3 and F4/80), image analysis was conducted using ImageJ software ver. 1.46 (National Institutes of Health, Bethesda, MD, USA). The number of cells positive for a specific host cell marker or signal intensities from various channels in 10–15 fields of fluorescently stained tissue sections were measured. Pooled average field intensity measured from at least 5 fields per slide containing minimum 100 cells per field was used for calculations. 

### 4.8. Statistical Analysis

Statistical analysis of data was performed using GraphPad Prism 9 (GraphPad Software, Boston, MA, USA). Comparisons between two experimental groups were performed using an unpaired *t*-test with Welsh correction, and for multiple group comparison, one-way ANOVA with Tukey’s correction or two-way ANOVA was used. For all experimental data comparisons between groups, a *p*-value of <0.05 was considered statistically significant. 

## 5. Conclusions

In conclusion, we have developed a novel humanized mouse model of pulmonary Mtb infection that shows a similar immunopathology as seen in patients with active pulmonary TB. This model can be a useful tool to study the host–pathogen interactions in mono- or co-infection settings, as well as in evaluating novel vaccine and drug candidates for TB and for other infectious diseases. However, long-term Mtb infection studies and deeper functional characterization of various immune cells and their role in TB pathogenesis are warranted before fully understanding and utilizing this humanized mouse model. 

## Figures and Tables

**Figure 1 ijms-25-01656-f001:**
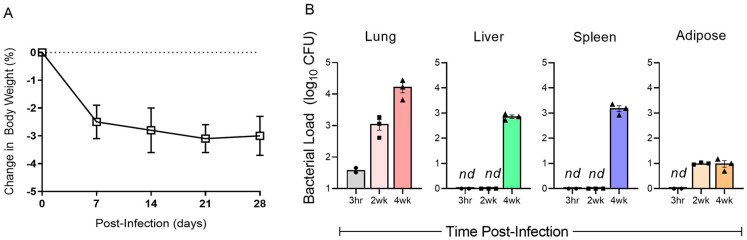
Body weight changes and bacterial burden in various tissues of Mtb-infected humanized mice. (**A**). Change in body weight shown as percentage reduction from the time of Mtb infection (T = 0) until 28 days (4 weeks) post infection. The body weight of uninfected control mice was used as normalizer and shown as a dotted line. Data shown are mean ± SD from n = 8 for T = 0, and n = 5 for 7 and 14 days and n = 3 for 28 days post infection. No significant weight loss was noted in animals without Mtb infection (n = 3) (**B**). Mtb load in the lungs, liver, spleen, and adipose tissue of humanized mice from the initial implantation (3 h), 2 weeks and 4 weeks post infection measured as number of bacterial CFU and presented in log scale. Data shown are mean ± SD from n = 2–3 per time point. *nd*—not detected.

**Figure 2 ijms-25-01656-f002:**
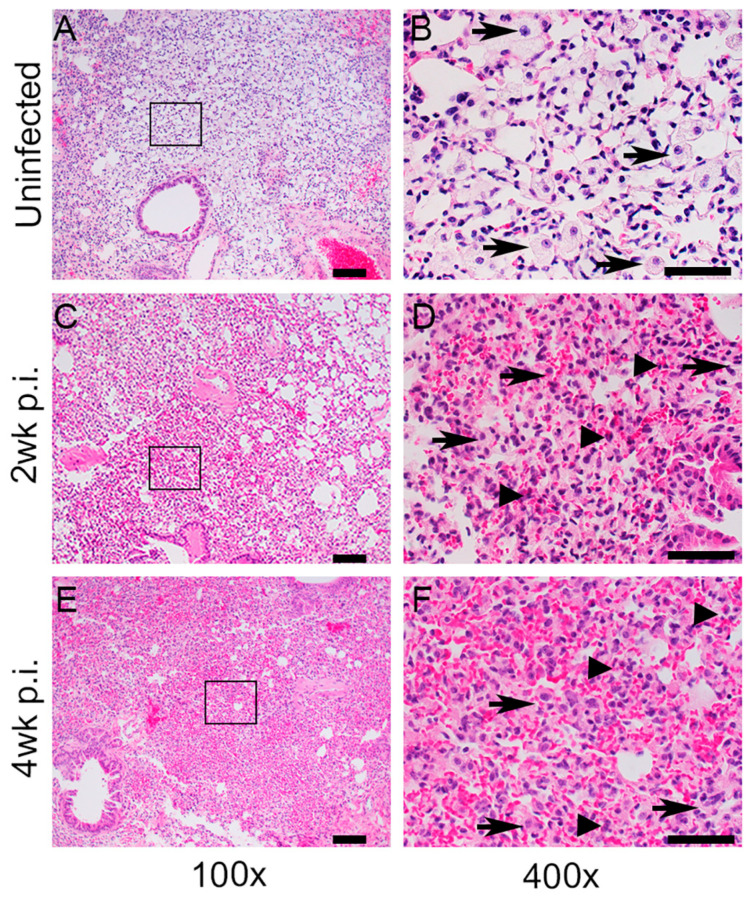
Histopathological analysis of lungs in humanized mice with or without Mtb infection. (**A**,**B**). Representative H&E-stained images of lung sections from uninfected humanized mice. The boxed region in (**A**) is magnified and shown in panel (**B**). Clearly visible interstitial macrophages (arrows in (**B**)) with no inflammation can be seen in the uninfected mouse lungs. (**C**,**D**). Representative H&E-stained images of lung sections from Mtb-infected humanized mice at 2 weeks post infection (2 wk p.i.). The boxed region in (**C**) is magnified and shown in panel (**D**). At this timepoint, a moderate level of inflammation and immune cell infiltration, including macrophages (arrows in (**D**)) and neutrophils (arrow heads in (**D**)), can be seen. (**E**,**F**). Representative H&E-stained images of lung sections from Mtb-infected humanized mice at 4 weeks post infection (4 wk p.i.). The boxed region in (**E**) is magnified and shown in panel (**F**). Severe inflammation and exacerbated immune cell infiltration, including macrophages (arrows in (**F**)) and neutrophils (arrow heads in (**F**)), can be seen at this time point. n = 2–3 per time point. Images (**A**,**C**,**E**) are shown at 100× magnification and (**B**,**D**,**F**) are shown at 400× magnification. The scale bar in (**A**,**C**,**E**) is 100 µm, and the scale bar in (**B**,**D**,**F**) is 50 µm.

**Figure 3 ijms-25-01656-f003:**
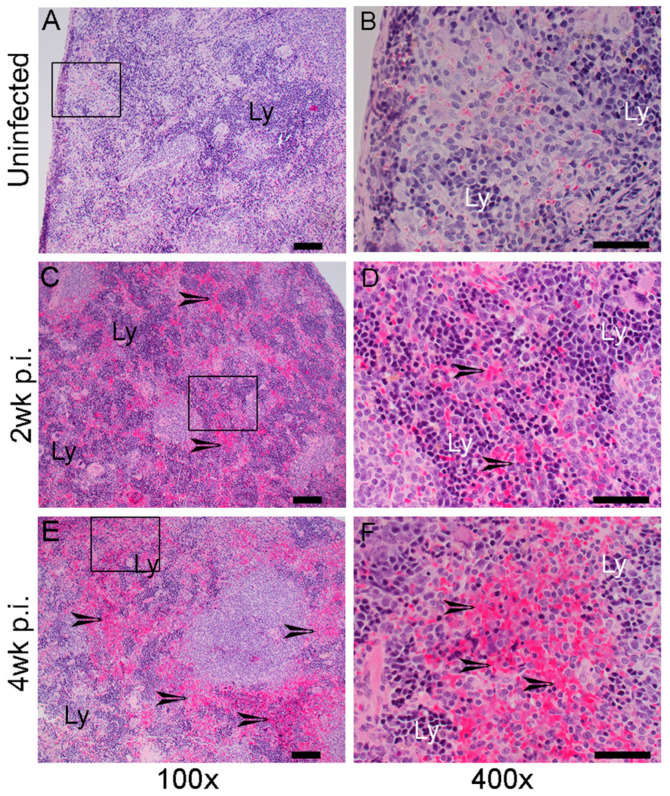
Histopathological analysis of the spleen in humanized mice with or without Mtb infection. (**A**,**B**). Representative H&E-stained images of spleen sections from uninfected humanized mice. The boxed region in (**A**) is magnified and shown in panel (**B**). A mild level of lymphocytic accumulation (**Ly**) can be seen in these animals. (**C**,**D**). Representative H&E-stained images of spleen sections in Mtb-infected humanized mice at 2 weeks post infection (2 wk p.i.). The boxed region in (**C**) is magnified and shown in panel (**D**). A moderate level of inflammation (arrows) and immune cell infiltration, particularly lymphocytes (**Ly**), can be seen. (**E**,**F**). Representative H&E-stained images of spleen sections in Mtb-infected humanized mice at 4 weeks post infection (4 wk p.i.). The boxed region in (**E**) is magnified and shown in panel (**F**). Severe inflammation (arrows) and foci of lymphocytes (**Ly**) can be seen at this time point. n = 2–3 per time point. Images (**A**,**C**,**E**) are shown at 100× magnification and (**B**,**D**,**F**) are shown at 400× magnification. The scale bar in (**A**,**C**,**E**) is 100 µm. The scale bar in (**B**,**D**,**F**) is 50 µm.

**Figure 4 ijms-25-01656-f004:**
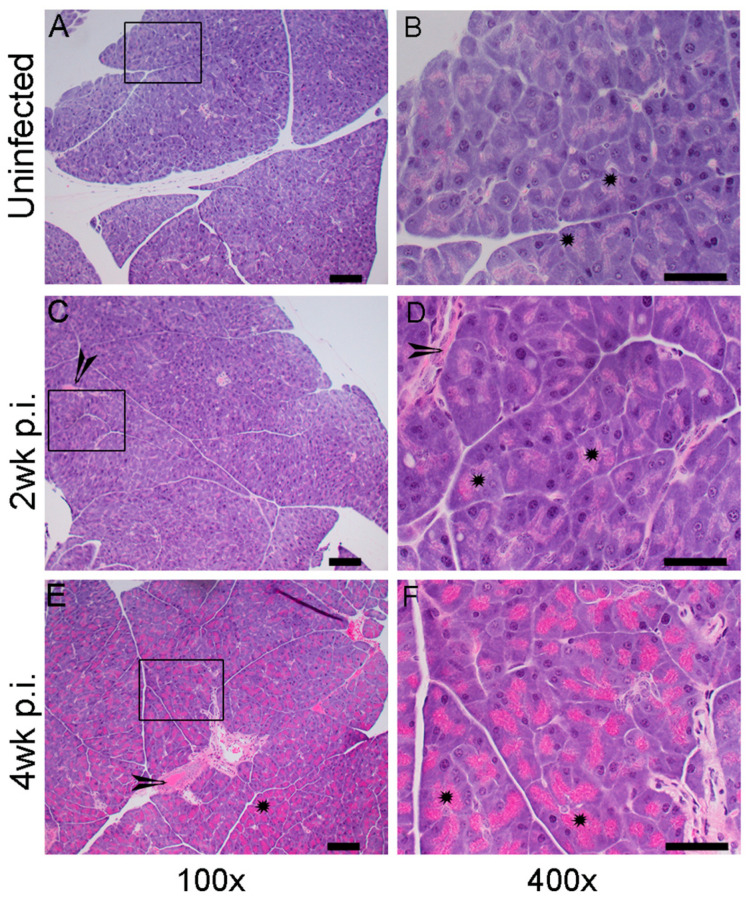
Histopathological analysis of adipose tissue in humanized mouse with or without Mtb infection. (**A**,**B**). Representative H&E-stained images of adipose tissue from uninfected humanized mice. The boxed region in (**A**) is magnified and shown in panel (**B**). Adipocytes with no inflammatory reaction was seen in these uninfected tissue (* in (**B**)) (**C**,**D**). Representative H&E-stained images of adipose tissue from Mtb-infected humanized mice at 2 weeks post infection (2 wk p.i.). The boxed region in (**C**) is magnified and shown in panel (**D**). A moderate level of inflammation and immune cell infiltration (arrow in (**C**,**D**) and * in (**D**) can be seen at this time point. (**E**,**F**). Representative H&E-stained images of adipose tissue in Mtb-infected humanized mice at 4 weeks post infection (4 wk p.i.). The boxed region in (**E**) is magnified and shown in panel (**F**). Severe inflammation and immune cell activation can be seen (arrow in (**E**) and * in (**E**,**F**)) at this timepoint. n = 2–3 per time point. Images (**A**,**C**,**E**) are shown at 100× magnification and (**B**,**D**,**F**) are shown at 400× magnification. The scale bar in (**A**,**C**,**E**) is 100 µm. The scale bar in (**B**,**D**,**F**) is 50 µm.

**Figure 5 ijms-25-01656-f005:**
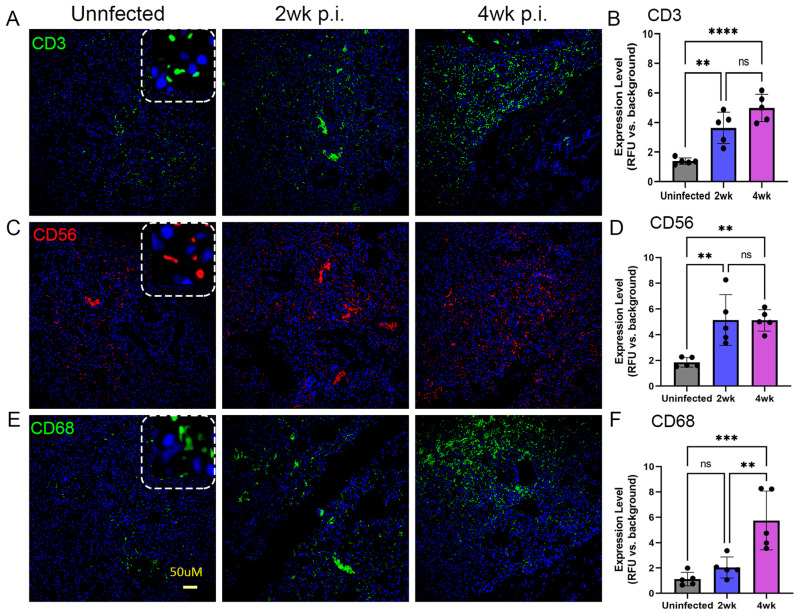
Spatial imaging analysis of T cells, NK cells, and monocytes/macrophages in the lungs of humanized mice with or without Mtb infection for 2 and 4 weeks post infection. Immunohistochemistry with immune cell-specific antibodies was used to determine the spatial distribution of CD3+ T cells (**A**), CD56+ NK cells (**C**), and CD68+ monocytes/macrophages (**E**) in the humanized mouse lungs without Mtb infection (**Left panels**), or at 2 weeks post infection (**Middle panels**) or 4 weeks post infection (**Right panels**). The dotted square inset in (**A**,**C**,**E**) shows a higher magnification to indicate the respective antibody staining pattern. (**B**) Determination of the frequency of CD3+ T cells in the uninfected and Mtb-infected humanized lungs at 2 or 4 weeks. (**D**) Determination of the frequency of CD56+ NK cells in the uninfected and Mtb-infected humanized lungs at 2 or 4 weeks. (**F**) Determination of the frequency of CD68+ monocytes/macrophages in the uninfected and Mtb-infected humanized lungs at 2 or 4 weeks. The scale bar in (**E**) (50 µm) is common for all other panels. Data shown are mean ± SE from 5 fields containing a minimum 100 cells/field. Data acquired from 2–3 animals per timepoint and analyzed by one-way ANOVA with Tukey’s post-hoc testing. ** *p* < 0.01; *** *p* < 0.005; **** *p* < 0.001; ns—not significant.

**Figure 6 ijms-25-01656-f006:**
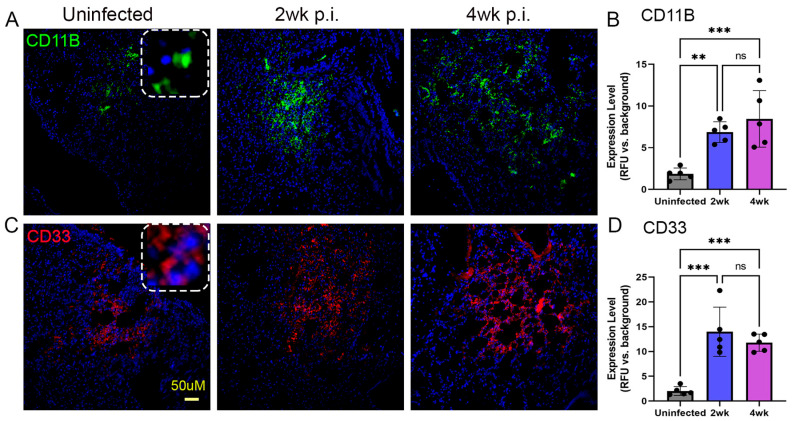
Spatial imaging analysis of MDSCs in the lungs of humanized mice with or without Mtb infection for 2 and 4 weeks post infection. CD11B (**A**) and CD33 (**C**) antibodies were used to determine the spatial distribution of MDSCs in the humanized mouse lungs without Mtb infection (**Left panels**), or at 2 weeks post infection (**Middle panels**) or 4 weeks post infection (**Right panels**). The dotted square inset in (**A**,**C**) shows a higher magnification to indicate the respective antibody staining pattern. (**B**) Determination of the frequency of CD11B+ MDSCs in the uninfected Mtb-infected humanized lungs at 2 or 4 weeks. (**D**) Determination of the frequency of CD33+ MDSCs in the uninfected and Mtb-infected humanized lungs at 2 or 4 weeks. The scale bar in (**C**) (50 µm) is common for all other panels. Data shown are mean ± SE from 5 fields containing a minimum 100 cells/field. Data acquired from 2–3 animals per timepoint and analyzed by one-way ANOVA with Tukey’s post-hoc testing. ** *p* < 0.01; *** *p* < 0.005; ns—not significant.

**Figure 7 ijms-25-01656-f007:**
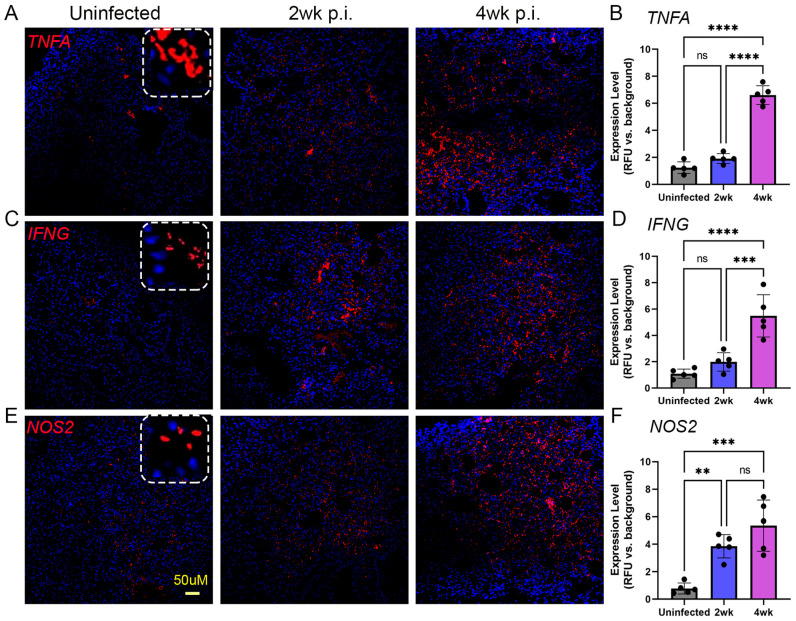
Spatial imaging analysis of TNFα, IFNγ, and NOS2 expressing cells in the lungs of humanized mice with or without Mtb infection for 2 and 4 weeks post infection. The mRNA-based single molecule fluorescent in situ hybridization (sm-FISH) technique was used to determine the spatial expression of TNFα, IFNγ, and NOS2 expressing cells in the humanized mouse lungs without Mtb infection (**Left panels**), or at 2 weeks post infection (**Middle panels**) or 4 weeks post infection (**Right panels**). The dotted square inset in (**A**,**C**,**E**) shows a higher magnification to indicate the respective smFISH staining pattern. (**B**) Expression level of *TNFA* in the uninfected and Mtb-infected humanized lungs at 2 or 4 weeks. (**D**) Expression level of *IFNG* in the uninfected and Mtb-infected humanized lungs at 2 or 4 weeks. (**F**) Expression level of *NOS2* in the uninfected and Mtb-infected humanized lungs at 2 or 4 weeks. The scale bar in (**E**) (50 µm) is common for all other panels. Data shown are mean ± SE from 5 fields containing a minimum 100 cells/field. Data acquired from 2–3 animals per timepoint and analyzed by one-way ANOVA with Tukey’s post-hoc testing. ** *p* < 0.01; *** *p* < 0.005; **** *p* < 0.001; ns—not significant.

**Figure 8 ijms-25-01656-f008:**
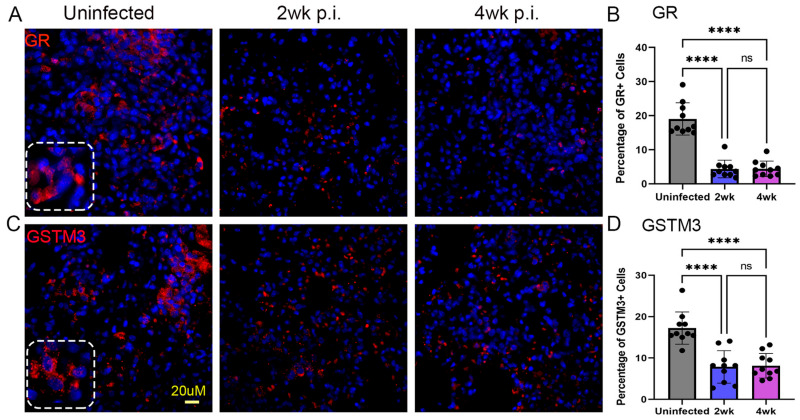
Spatial imaging analysis of glutathione reductase (GR) and glutathione S transferase (GSTM3) expressing cells in the lungs of humanized mice with or without Mtb infection for 2 and 4 weeks post infection. An antibody specific to GR or GSTM3 was used to determine the spatial localization of cells expressing these markers in the humanized mouse lungs without Mtb infection (**Left panels**), or at 2 weeks post infection (**Middle panels**) or 4 weeks post infection (**Right panels**). The dotted square inset in (**A**,**C**) shows a higher magnification to indicate the respective antibody staining pattern. (**B**) Percent of **GR**-positive cells in the uninfected and Mtb-infected humanized lungs at 2 or 4 weeks. (**D**) Percent of **GSTM3**-positive cells in the uninfected and Mtb-infected humanized lungs at 2 or 4 weeks. The scale bar in (**C**) (20 µm) is common for all other panels. Data shown are mean ± SE from 10 fields containing a minimum 50 cells/field. Data acquired from 2–3 animals per timepoint and analyzed by one-way ANOVA with Tukey’s post-hoc testing. **** *p* < 0.001; ns—not significant.

## Data Availability

The data supporting reported results can be obtained from the corresponding author (S.S.) upon formal requisition.

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
