# Peer review of "Immunopathology of Pulmonary Mycobacterium tuberculosis Infection in a Humanized Mouse Model"

_ijms, 2024, doi:10.3390/ijms25031656_

Round 1

Reviewer 1 Report

Comments and Suggestions for Authors

Mycobacterium tuberculosis is one of the leading causative agents for death infected by single pathogen. A less costly animal model that better mimic human disease is critical in research and clinical trials to control the infection. In the manuscript submitted by Kolloli, et al., the authors described a novel humanized mouse model which is used for study infection of M. tuberculosis. Multiple approaches were used to determine the characteristics of the mouse model during infection. The results are interesting. Better clarity in writing will further improve the quality of the manuscript.

Major comments

1.     L76-86. Mouse TB model in humanized mice should be introduced in more details.  

2.     Experiments of GSH and GSTM3 were not described in Methods.

3.     L200-214. Since the humanization of the mice model is novel as described, how many animals were used for the conclusion of life span? Were the various cell types detected in the current animal model or just reported in the referred papers?

4.     Fig. 1. and L241. “n=5 per time point” – I assume that the mice at each time point were from the same group. In Figure 1A, clarify how many mice were used as uninfected controls and how many were infected. How were the control animals used to normalize the weight loss of the infected mice? Is it better just show the weight changes of the two groups in percentage of day 0? The current data cannot show the weight changes of the uninfected humanized mice.

5.     Figure legends of supplemental figures were not provided.

Minor comments

1.     L41. LTBI – latent TB infection

2.     L44-45. Either “increases the risk of mortality” or “increases the mortality rate”

3.     L49. Rewrite this sentence.

4.     L77. Define NSG mice (Ref L113).

5.     L95. Add “M3” after S-transferase.

6.     L109. Incubating H37Rv strain on supplemented 7H10 plates for 4-6 weeks seems long. Was there a reason to keep 6 weeks?

7.     L128-142. What time point after humanization were the mice used for infection?

8.     L202. Clarify “This approach” as multiple approaches were listed.

9.     L542. Organization WH? (WHO)

10.  Fig. 1B. Is it possible to show individual mouse in the graph, especially the sample number is small?

11.  Fig. 2-4. How many mice were used for each experiment?

12.  Change Anova to ANOVA in figure legends.

Comments on the Quality of English Language

The manuscript is overall well written. For comments on the language, please see Comments and Suggestions for Authors.

Author Response

Major comments

Comment: L76-86. Mouse TB model in humanized mice should be introduced in more details.  

Response: As suggested by the reviewer, we have elaborated more on the usage of humanized mouse Tb models in the introduction section.

Comment: Experiments of GSH and GSTM3 were not described in Methods.

Response: The experimental conditions, including GSTM3 and GR antibody information is already present in the Methods section 2.5.

Comment: L200-214. Since the humanization of the mice model is novel as described, how many animals were used for the conclusion of life span? Were the various cell types detected in the current animal model or just reported in the referred papers?

Response: The reviewer’s perception was correct. The life span and various cell types of the humanized mice described in this manuscript were reported in previous publication (Ref# 25), which is duly cited in our article.

Comment: Fig. 1. and L241. “n=5 per time point” – I assume that the mice at each time point were from the same group. In Figure 1A, clarify how many mice were used as uninfected controls and how many were infected. How were the control animals used to normalize the weight loss of the infected mice? Is it better just show the weight changes of the two groups in percentage of day 0? The current data cannot show the weight changes of the uninfected humanized mice.

Response:  The data in Figure 1A was taken from the same group of mice at each time point. n=8 for T=0, and n=5 for 7 and 14 days and n=3 for 28 days post infection. Since there was no significant weight loss in the uninfected animals (n=3), we used that as normalizer to highlight the weight loss caused by TB in the infected animals in Figure 1A. This information is added in Figure 1 legend of the revised manuscript. 

Comment: Figure legends of supplemental figures were not provided.

Response: Figure legends of supplemental figures are added in the revised manuscript.

Minor comments 

  1. LTBI – latent TB infection.

       Corrected.

  1. L44-45. Either “increases the risk of mortality” or “increases the mortality rate”

       Corrected.

  1. Rewrite this sentence.

        Corrected.

  1. Define NSG mice (Ref L113).

        Corrected.

  1. Add “M3” after S-transferase.

        Corrected.

  1. Incubating H37Rv strain on supplemented 7H10 plates for 4-6 weeks seems long. Was there a reason to keep 6 weeks?

        We count the colonies every week from 4 to 6 weeks.  The longer incubation is needed to ensure that if any slow-growing colonies emerge, particularly when tissue lysates were plated.  

  1. L128-142. What time point after humanization were the mice used for infection?

         After 8 weeks of reconstitution, the humanized mice were used for infection studies. This information is added in the Methods section 2.2.

  1. Clarify “This approach” as multiple approaches were listed.

         The sentence is reframed as “. This approach of tissue engraftment followed by cytokine treatment is novel….” In the Results section 3.1.

  1. L542. Organization WH? (WHO)

         Corrected

  1. Fig. 1B. Is it possible to show individual mouse in the graph, especially the sample number is small?

         As suggested by the reviewer, we have changed Fig 1B to show individual data points on the graph.

  1. Fig. 2-4. How many mice were used for each experiment?

          For Figure 2-4, n=2-3 per time point.  This information is added in the legends of Figures 2-4.

  1. Change Anova to ANOVA in figure legends.

          Corrected

Reviewer 2 Report

Comments and Suggestions for Authors

1. Nowadays, it is strongly recommended by ethical committees to include an equal number of male and female mice in experiments. This ensures that all the data obtained can be analyzed separately for each sex. Have you taken into account the number of male and female mice in the experiment and maintained a 1:1 ratio? Additionally, have you considered using non-humanized mice, such as C57BL/6, as a control group?

2. In Line 238, Figure 1, the manuscript utilizes a histogram to count the bacterial load in different tissues. Would a scatter plot be more accurate in reflecting the bacterial load in each individual mouse across different tissues?

3. In a separate research report, it was found that using humanized mice and infecting them with 20 cfu of MTB, significant pathological changes similar to human caseous necrotic granulomas were observed in the mouse lung tissue after 35 days. However, this experiment is not depicted in Figure 2 at Line 265. Could such a significant change be attributed to the dosage of MTB infection and the relatively short observation period of four weeks?

Author Response

Comment-1. Nowadays, it is strongly recommended by ethical committees to include an equal number of male and female mice in experiments. This ensures that all the data obtained can be analyzed separately for each sex. Have you taken into account the number of male and female mice in the experiment and maintained a 1:1 ratio? Additionally, have you considered using non-humanized mice, such as C57BL/6, as a control group?

Response:  We fully agree with the view of this reviewer regarding animal usage, and we apologize for not including animal from both sexes.  Due to the complexity involved in creating these humanized mice, including availability of human tissues, surgical implantations, post-surgical monitoring etc, and the high cost of producing the humanized mice, we couldn’t include animals of both sexes.  Further, in this article we are not evaluating the data to determine the effect of biological sex on tuberculosis. In the future, we will consider conducting a detailed study involving 1:1 ratio of male and female mice in experiments as suggested by the reviewer. Although it is a good idea to compare the non-humanized mice models, such as C57BL/6 with the humanized mice, these two models are fundamentally different and we believe that the humanized mice can be presented as a stand-alone study.  In the future, we will consider conducting a detailed study comparing the humanized and non-humanized mice models as suggested by the reviewer

Comment-2. In Line 238, Figure 1, the manuscript utilizes a histogram to count the bacterial load in different tissues. Would a scatter plot be more accurate in reflecting the bacterial load in each individual mouse across different tissues?

Response: As suggested by both the reviewers, we have now changed Fig 1B to show individual data points on the graph.

Comment-3. In a separate research report, it was found that using humanized mice and infecting them with 20 cfu of MTB, significant pathological changes similar to human caseous necrotic granulomas were observed in the mouse lung tissue after 35 days. However, this experiment is not depicted in Figure 2 at Line 265. Could such a significant change be attributed to the dosage of MTB infection and the relatively short observation period of four weeks?

Response: As the reviewer correctly pointed out, very few studies have used low number of Mtb to infect humanized mice (cited in Ref.15-23).  Some of these reports show necrotic lesions as early as 4 weeks post infection (Ref#17), similar to our observation in this manuscript. In some studies, the Mtb infected HIS-NSG mice showed severe signs of disease at 25 days and succumbed to infection by 33 to 35 days post infection (Ref#16).  However, our current study did not include these time points (35 days). In Figure-2, we showed severe inflammation and exacerbated immune cell infiltration at 4 weeks post infection, which is consistent with progressive disease in our humanized model.  Our observations are in-line with several other studies as we cited in Ref 15-23. In fact, as the reviewer has mentioned the difference in granuloma pathology is affected by multiple factors, including the initial inoculum, the nature of Mtb strain used, the route of infection, the genetic background of mice and the time of analysis of samples.